# Strategies to resolve the gap in adolescent tuberculosis care at four health facilities in Uganda: The teenager's TB pilot project

Samson Omongot[1,2]*, Winters Muttamba[1,3], Irene Najjingo[1], Joseph Baruch Baluku[1,4], Sabrina Kitaka[5], Stavia Turyahabwe[6], Bruce Kirenga[1,2]

1 Makerere University Lung Institute, Makerere University, Kampala, Uganda, 2 Department of Medicine, Makerere University College of Health Sciences, Kampala, Uganda, 3 Division of Infection and Global Health, School of Medicine, University of St Andrews, St. Andrews, United Kingdom, 4 Division of Pulmonology, Kiruddu National Referral Hospital, Kampala, Uganda, 5 Department of Pediatrics, Makerere University College of Health Sciences, Kampala, Uganda, 6 National Tuberculosis and leprosy Programme, Ministry of Health (MOH), Kampala, Uganda

* somongot@gmail.com

**Data Availability Statement:** All relevant data are within the manuscript and its Supporting information files.

## Abstract

In 2021, an estimated 10.6 million people fell ill with tuberculosis (TB) globally and 11.3% were children. About 40% of children aged five to fourteen years with TB are missed annually. In Uganda, 44% of adolescents with chronic cough of more than two weeks do not seek care from health facilities. Therefore, strategies to promote health care-seeking behaviour among adolescents were urgently needed to resolve the gap. In regard to this, the research project utilized a before and after design, in which the number of adolescents (10-19years) enrolled in the project health facilities were compared before and after the intervention. The intervention package that comprised of tuberculosis awareness and screening information was developed together with adolescents, thus; a human-centred approach was used. The package consisted of TB screening cards, poster messages and a local song. The song was broadcasted in the community radios. Poster messages were deployed in the community by the village health teams (VHTS). The TB screening cards were given to TB positive and presumptive adults to screen adolescents at home. Adolescents that were found with TB symptoms were referred to the project health facilities. Socio-demographic and clinical characteristics of eligible adolescents were collected in a period of six months from Kawolo, Iganga, Gombe and Kiwoko health facilities. To determine the effectiveness of the package, before and after intervention data were equally collected. A total of 394 adolescents were enrolled, majority (76%) were school going. The intervention improved adolescent TB care seeking in the four project health facilities. The average number of adolescents screened increased from 159 to 309 (incidence rate ratio (IRR) = 1.9, P<0.001, 95% CI [1.9, 2.0]). Those presumed to have TB increased from 13 to 29(IRR = 2.2, P<0.001, 95% CI [1.9, 2.5]). The ones tested with GeneXpert increased in average from 8 to 28(IRR = 3.3, P<0.001, 95% CI [2.8, 3.8]). There was a minimal increase in the average monthly number of adolescents with a positive result of 0.8, from 1.6 to 2.4(p = 0.170) and linkage to TB care services of 1.1, from 2 to 3.1(p = 0.154). The project improved uptake of TB services among

**Funding:** Funding was from government of Uganda under Makerere University research and innovation fund. The funders had no role in study design, data collection and analysis, decision to publish, or preparation of the manuscript.

**Competing interests:** The author(s) declare that they have no competing interests.

adolescents along the TB care cascade. We recommend a robust and fully powered randomized controlled trial to evaluate the effectiveness of the Package.

## Background

Globally, an estimated 10.6 million people fell ill with tuberculosis (TB) in 2021, an increase of 4.5% from the 10.1 million TB cases diagnosed in 2020 when TB screening declined due to COVID-19 outbreak and restrictions [1] Among the 10.6 million people estimated, 1.2 million were children with an estimated TB burden of 11% [1]. Globally, in 2021, the number of Incident TB cases among female and males aged between five to fourteen years were 306,000 and 299,000 per 100,000 populations respectively. Nonetheless, the number of females and males in the same age group notified were 146,034 and 132,652 respectively [1]. Africa in that year had an estimated TB incident rate of 81,000 females and 80,000 males, with the number of cases notified among female and male being 40, 771 and 40,966 respectively [1]. In addition, Uganda in the same year reported TB incidence of 199,000, and notification of 163,000 cases per 100,000 populations [1]. About 40% of TB cases among children aged between five and fourteen years were missed annually in health centres, irrespective of the global prevalence to notification ratio of 1.7 [2]. Additionally, there is paucity of epidemiological data to guide interventions that address the specific needs of adolescents [3]. One of the barriers to TB case finding in adolescents is low rates of TB screening and testing in this age group, this was demonstrated in the Uganda National TB survey which revealed that only 56% of adolescents reporting a chronic cough of $\geq$ two weeks sought care from health facilities [4]. Although there are several other reasons for the low TB testing in adolescents; lack of skills and knowledge by the health care workers to deliver adolescent friendly health services are some of the leading barriers [5]. Studies have revealed that delays in diagnosis, stigma related with diagnosis and treatment, long waiting hours at health facilities, absence of nutritional support for patients with TB, and absence of comprehensive psychosocial support programs are barriers to access and adherence to TB care [5, 6]. Strategies including community engagement, training health workers and strengthening public-private partnerships have been found vital in TB control and reducing the missed cases [7]. Despite these generalised findings, strategies to promote care-seeking behaviours among adolescents and ensuring adequate evaluation in health facilities are needed to bridge the gap in the quality of TB care among adolescents. In response to this, we piloted an adolescent TB care package at four health facilities aimed at improving adolescent TB care seeking behaviour.

## Methods

### Project settings

The project sites included district level hospitals that were purposively selected. These were health facilities with TB diagnostic and treatment units located in both rural and urban places in the central region of Uganda. Among the four health facilities selected, Kiwoko and Gombe hospitals were rural, while Iganga and Kawolo hospitals were urban. Gombe is a public health facility with 100 bed capacity, while Kiwoko is a faith- based private health facility with 204 bed capacity. The other two facilities (Iganga and Kawolo hospitals) are both public urban health facilities with up to 100 bed capacity. All the health facilities offer TB and other services, such as out-patients, in-patients, antenatal, HIV, eye, dental, nutrition and community services.

## Project design

This interventional research project utilized a before and after design, in which the number of adolescents enrolled in the project health facilities were compared before and after the intervention. The intervention package that comprised of tuberculosis awareness and screening information was developed together with adolescents, thus; a human-centered approach was used. The package contained adolescent TB educative and informative messages simulated in form of educational posters, TB awareness local song ("*Bulamu bwo*") and TB screening cards. This package was implemented in four project health facilities. Implementation phase had a number of activities that were kick started by a meeting with relevant stakeholders (Ministry of Health, educational and political leaders, and key adolescent health care providers). The meeting was led by the National TB programme, and the research team provided technical support. During this meeting, the project team presented data on the clinical and economic burden and unmet needs of TB among adolescents. Besides different stakeholders being engaged, health workers were given three day training by TB programme staff, assisted by the research team.

The training focused on four key ingredients of adolescent friendly health services such as being non-judgmental, friendly adolescent services, service demand creation and community awareness/ support.

Implementation of TB awareness package was done through the support of project volunteers, research assistants, village health teams (VHTs) and health workers. The VHTs displayed posters on collection points of adolescents in the communities, in the project health facilities and drug shops/stores to sensitize adolescents to seek TB care. The VHTs also identified and engaged local radios serving respective communities to broadcast the song. The health workers and volunteers oriented the adults who sought care in the TB clinics or with TB symptoms and had adolescents at home on filling of the TB- screening cards. The adults were eventually given the cards to screen the adolescents at home, they were asked to send adolescents found with TB symptoms to the project health facilities to seek care. Poster messages and screening cards were implemented for six months (October, 2021 to March, 2022). The local song was broadcasted on the radio stations for three months (January to March 2022). Adolescents who reported to the health facility from October 2021 to March 2022 were subjected to TB screening. Those who presented with predefined tuberculosis symptoms according to the World Health Organization's (WHO) criteria (cough for $\geq 2$ weeks; persistent fever for $\geq 2$ weeks; noticeable unintentional weight loss; and excessive night sweats), were consented if they were eighteen years or more of age, or assented if less than eighteen years. All those who consented or assented were enrolled into the project.

These adolescents were taken through the entire TB care cascade (screening, testing and linkage). All responses on socio-demographic and clinical characteristics of the enrolled adolescents were captured into a tablet computer and eventually sent to the server. The data was retrieved from the server, cleaned and analyzed. To assess the impact of the package, before and after intervention data on adolescents' screened, presumed, tested, TB positives and those linked to TB care was collected from the health facility registers and analyzed.

## Project participants

The project participants included adolescents who presented to the health facilities at different service delivery points such as outpatient department (OPD), Anti-retroviral therapy (ART) clinic, maternal and reproductive health (MRH) clinic, inpatient wards and TB clinics. We included all adolescents who presented with at least one symptom suggestive of TB; predefined according to the World Health Organisation's (WHO) criteria (cough for $\geq 2$ weeks; persistent

fever for $\geq 2$ weeks; noticeable unintentional weight loss; and excessive night sweats). TB patients who were already on TB treatment or had been screened in the same project within two weeks were excluded.

## Data quality control

Before data collection was initiated in the project health facilities, electronic data collection tools, informed consent and assent forms were designed by the project investigators. The tools and other research documents were approved by the ethics research committee. Health workers, research assistants, research volunteers and village health teams (VHTs), were trained on the project protocol and supervised during data collection. The tools were pre-tested before data collection. Electronic data were sent to the server on a daily basis.

## Data collection

Adolescents who came to the research project sites during the intervention period were identified from the service delivery points (OPD, ART clinic, MRH clinic, TB clinic and in-patient wards) and screened for TB using the national screening algorithm. Those who had at least one of the TB symptoms and not on anti-TB treatment were considered eligible. All eligible adolescents who were eighteen years and above ($\geq 18$ years) were consented before enrolment. The ones who were less than eighteen years ($<18$ years) were assented for by their parents or health workers if they had no caregivers. All the ones who were consented or assented were provided with identification numbers and enrolled into the project. Socio-demographics, clinical characteristics and aggregate level data were collected using kobo collect electronic software. Data were kept in a server prior to analysis.

## Data management and analysis

The data were extracted from the server, cleaned and exported to STATA version 14 for analysis. Descriptive statistics were used to summarize data into proportions and frequencies for categorical variables, and means and median for continuous variables depending on the distribution. An independent sample T-test was used to compare differences between the means. Logistic regression model was used to determine the incident rate ration (IRR). Statistically significant associations were rated at p-value of less than 0.05 at 95% confidence interval (CI).

## Ethical considerations

The research project ethical approvals were obtained from Mulago Hospital Research.

Ethics Committee (MHREC 1922) and the Uganda National Council of Science and Technology (UNCST HS1042ES). Administrative approval (ADM. 105/309/05) was obtained from Ministry of health. All participants eighteen years and above, gave written informed consents, while adult caregivers or health workers (in absence of caregivers) gave written informed assents for participants less than eighteen years of age.

## Results

### Socio- demographic characteristics

Data on socio-demographics was collected during the intervention phase to assess the response to the implemented package in regard to the different characteristics, such as: age, sex, employment, marital status, educational level and study site.

Three hundred ninety four (394) adolescents were enrolled. Majority, 198 (50%) were aged between 10–15 years. They were mainly females 255(65%), and still in school 298(76%). The

**Table 1. Socio-demographic characteristics.**

| Characteristics | Items | Frequency | Percentage (%) |
|---|---|---|---|
| Age | 10–15 years | 198 | 50.3 |
| | 16–19 years | 196 | 49.7 |
| Sex | Male | 139 | 35.3 |
| | Female | 255 | 64.7 |
| Employment status | Student | 298 | 75.6 |
| | Employed | 30 | 7.6 |
| | Unemployed | 66 | 16.8 |
| Marital status | Married | 58 | 14.7 |
| | Single | 336 | 85.3 |
| Educational Level | Complete primary | 23 | 5.8 |
| | Incomplete primary | 176 | 44.7 |
| | Complete secondary | 28 | 7.1 |
| | Incomplete secondary | 145 | 36.8 |
| | Tertiary | 5 | 1.3 |
| | Never attended school | 17 | 4.3 |
| Site | Kawolo Hospital | 137 | 34.8 |
| | Iganga Hospital | 88 | 22.3 |
| | Kiwoko Hospital | 77 | 19.5 |
| | Gombe Hospital | 92 | 23.4 |

majority, 336(85%) were unmarried. Kawolo hospital enrolled the highest number, 137(35%) of adolescents compared to the other health facilities as shown in Table 1 below.

## Clinical characteristics

Data on clinical characteristics was collected to find additional knowledge on possible sources of information and TB/HIV awareness by the adolescents as indicated below: Sixty four (16%) of the adolescents had never heard about TB. Seventy two (18%) reported a history of TB contact. Forty nine (68%) of the adolescents reported to have had TB contact with family members. One hundred twenty nine (33%) adolescents did not know their HIV status. Among those who knew their HIV status, sixty (15%) reported HIV positive, and fifty nine (98%) of the positives were on ART. Regarding the risk factors, seven (2%) had ever smoked, and thirty seven (9%) had ever consumed alcohol. Sixty eight (17%) had never heard about adolescent TB from any source. Among the one hundred twenty eight (32%) who reported to have got information from other sources, 9% received TB information from the local song that was broadcasted on the radios, 5% got TB information from the poster messages in the community and 1% were presumed referrals from the community identified by adults using the screening cards. All the above information is displayed in Table 2 below.

## Impact of package on TB care cascade

**Screening for TB.** There was an increase in the average numbers of adolescents screened for TB after the intervention across all the four project health facilities. The overall average increase in all the four health facilities was 150(from 159 to 309) adolescents screened. This increase resulted in 1.9 incident rate ratio (IRR) at 95% CI [1.9, 2.0] and was statistically significant ($p < 0.001$). Iganga hospital had a threefold increase in the average number screened of 416(from 213 to 629). This increase gave rise to 3.0 IRR at 95% CI [2.8, 3.2], statistically

**Table 2. Clinical characteristics.**

| Characteristic | Item | Frequency | Percentage |
|---|---|---|---|
| Do you know about TB? | Yes | 330 | 83.8 |
| | No | 64 | 16.2 |
| Close contact with someone with TB or chronic cough? | Yes | 72 | 18.3 |
| | No | 322 | 81.7 |
| If yes, relationship with contact? | Family members | 49 | 68.1 |
| | Family friend | 11 | 15.3 |
| | Fellow students | 12 | 16.7 |
| Do you know your HIV status? | Yes positive | 60 | 15.2 |
| | Yes negative | 205 | 52.0 |
| | I don't know | 129 | 32.7 |
| If yes and positive, are you on ART? | Yes | 59 | 98.3 |
| | No | 1 | 1.7 |
| Do you smoke tobacco? | Yes | 7 | 1.8 |
| | No | 387 | 98.2 |
| Do you take alcohol? | Yes | 37 | 9.4 |
| | No | 357 | 90.6 |
| Have you taken an x-ray? | Yes | 57 | 14.5 |
| | No | 337 | 85.5 |
| Have you collected sputum? | Yes | 305 | 77.4 |
| | No | 89 | 22.6 |
| Have you heard about adolescent TB? | Yes | 326 | 82.7 |
| | No | 68 | 17.3 |
| If yes, source of information about adolescent TB? | TEEN- TB Team | 266 | 37.6 |
| | Radio/TV | 138 | 19.5 |
| | Friend(s) | 83 | 11.7 |
| | VHTs | 67 | 9.5 |
| | Local song | 65 | 9.2 |
| | Social media | 46 | 6.5 |
| | Posters | 33 | 4.7 |
| | Screening cards | 9 | 1.3 |

significant (p < 0.001). Kawolo hospital had an increase of 84(from 185 to 269), resulting in 1.5 IRR at 95% CI [1.4, 1.6], and statistically significant (p < 0.001). Kiwoko Hospital had an increase of 12(from 15–27), resulting in 1.8 IRR at 95% CI [1.4, 2.4], the increase was statistically significant (p <0.001). Gombe hospital had an increase of 85(from 224 to 309), giving rise to 1.4 IRR at 95% CI [1.3–1.5], and this was statistically significant (p <0.001). The details of numbers screened before and after the intervention are indicated in Figs 1 and 2, and Table 3.

**TB presumptive patients.** There was an increase in the average number of presumptive TB adolescents identified during the project period. The overall average increase after the intervention was 16(from 13 to 29), thus; more than double the number before. This resulted in 2.2 incident rate ratio (IRR) at 95% CI [1.9, 2.5] that was statistically significant (p <0.001). Iganga hospital had more than double increase of 25(from 19 to 44). The increase gave rise to an IRR of 2.3 at 95% CI [1.9, 2.9], and this was statistically significant (p < 0.001). Kawolo hospital had a threefold increase of 35(from 18 to 53), resulting in an IRR of 2.9 at 95% CI [2.3–3.6], this was statistically significant (p < 0.001). Kiwoko hospital had an increase of 2, from 4

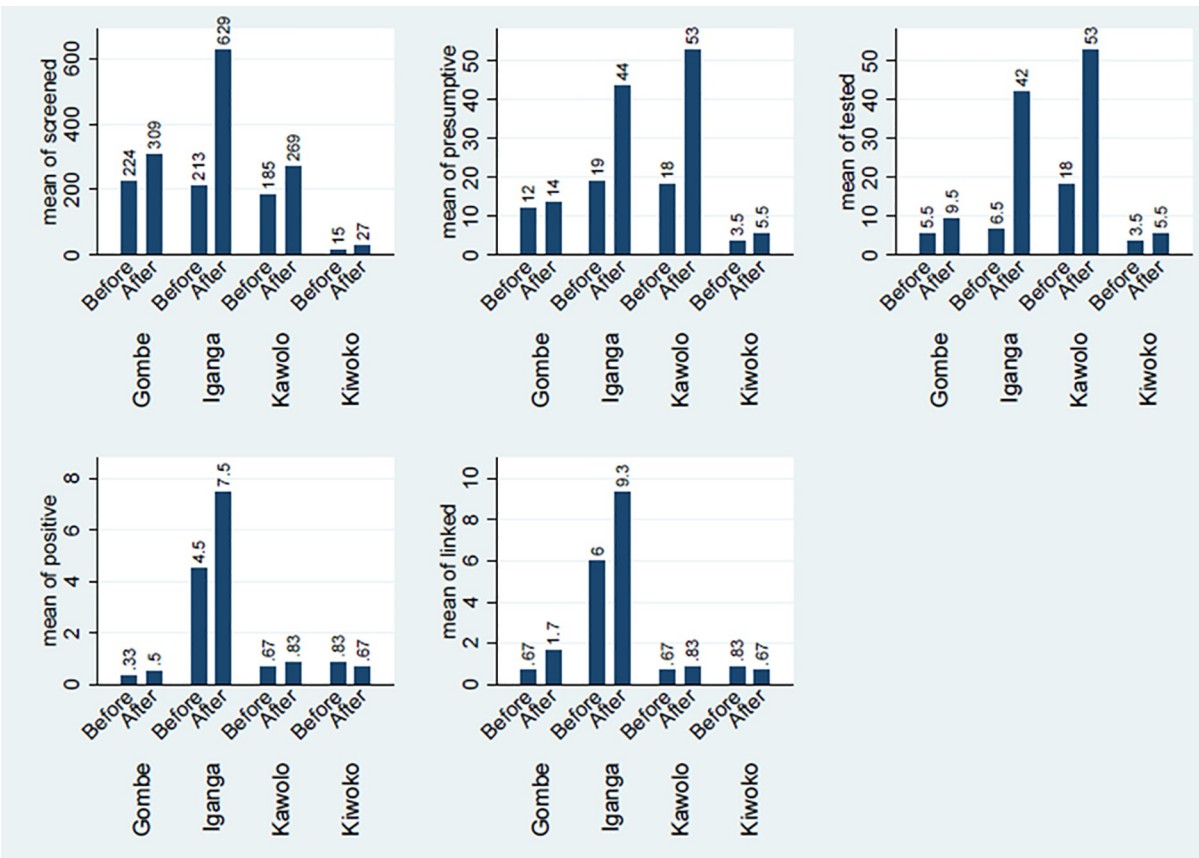

**Fig 1. Average number of adolescents across TB care cascade before & after intervention per health facility.**

to 6 resulting in an IRR of 1.6 at 95% CI [0.9, 5.4], an increase that was not statistically significant (p = 0.105). Gombe hospital had an increase of 2, from 12 to 14, giving rise to an IRR of 1.1 at 95% CI [0.8, 1.6], this was not statistically significant (p = 0.467). The details of the presumptive TB before and after the intervention are indicated in Figs 1 and 2, and Table 4.

**TB testing.** The overall average number of adolescents tested for TB increased by 10 (from 8 to 28) after the intervention, this was more than threefold increase resulting in an IRR of 3.3 at 95% CI[2.8,3.8]. This increase was statistically significant (p <0.001). The increase was more than six fold in Iganga hospital; 35(from 7 to 42), giving rise to an IRR of 6.5 at 95% CI [4.6–9.1], which was statistically significant (p <0.001). Kawolo hospital had a threefold increase of 35(from 18 to 53) resulting in an IRR of 2.9 at 95% CI [2.3, 3.6], and was statistically significant (p <0.001). Gombe hospital had an increase of 4(from 6 to 10), giving rise to an IRR of 1.7 at 95% CI [1.1, 2.7], which was statistically significant (p = 0.012). Kiwoko hospital had an increase of 2(from 4 to 6), resulting in an IRR of 1.6 at 95% CI [0.9, 2.7, this was not statistically significant (p = 0.105). The numbers tested for TB are indicated in Figs 1 and 2, and Table 5.

**TB positive adolescents.** Overall, there was a minimal average increase in the number of TB positive adolescents identified after the intervention of 1(from 2 to 3), and this was not statistically significant (p = 0.170). An increase of 3(from 5 to 8) was registered in Iganga hospital. An increase of 1(from 4 to 5) was registered in Gombe hospital. Kawolo hospital had an increase of 16(from 67 to 83). Kiwoko hospital registered a reduction of -16(from 0.83 to 0.67) as indicated in Figs 1 and 2.

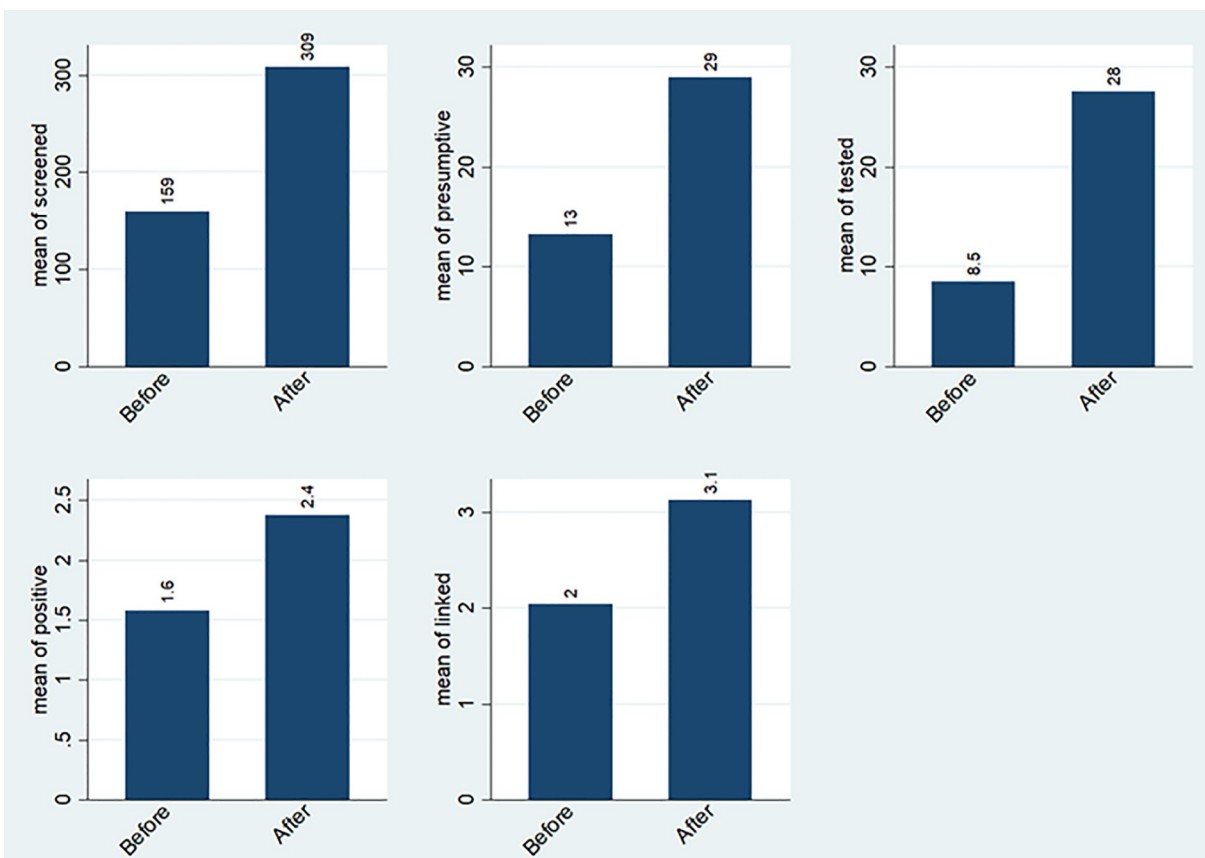

**Fig 2. Overall average number of adolescents across TB care cascade before & after the intervention.**

**Linkage to TB care and treatment.** There was a minimal improvement in linkage of adolescents to treatment and care. An average increase of 1(from 2 to 3) in the number of adolescents linked to care was achieved after the intervention. The increase was not statistically significant (p = 0.154). Gombe hospital had more than two-fold increase of 103(from 67 to

**Table 3. Incidence rate ratio (IRR) of adolescents screened for TB.**

| Facility | Period | Mean* | | Likelihood | | |
| --- | --- | --- | --- | --- | --- | --- |
| | | Mean(#screened) | p-value | IRR | p-value | 95% CI |
| **Overall** | Before | 159 | | | | |
| | After | 309 | 0.025 | **1.9** | <0.001 | (1.9–2.0) |
| Gombe Hospital | Before | 224 | | | | |
| | After | 309 | 0.190 | 1.4 | <0.001 | (1.3–1.5) |
| Iganga Hospital | Before | 213 | | | | |
| | After | 629 | 0.040 | 3.0 | <0.001 | (2.8–3.2) |
| Kawolo Hospital | Before | 185 | | | | |
| | After | 269 | 0.123 | 1.5 | <0.001 | (1.4–1.6) |
| Kiwoko Hospital | Before | 15 | | | | |
| | After | 27 | 0.071 | 1.8 | <0.001 | (1.4–2.4) |

* Independent samples t-Test

**Table 4. Incidence rate ratio (IRR) for TB presumptive adolescents.**

| Facility | Period | Mean* | | Likelihood | | |
|---|---|---|---|---|---|---|
| | | Mean(# presumptive) | p-value | IRR | p-value | 95% CI |
| **Overall** | Before | 13 | | | | |
| | After | 29 | 0.019 | **2.2** | <0.001 | (1.9–2.5) |
| Gombe Hospital | Before | 12 | | | | |
| | After | 14 | 0.356 | 1.1 | 0.467 | (0.8–1.6) |
| Iganga Hospital | Before | 19 | | | | |
| | After | 44 | 0.147 | 2.3 | <0.001 | (1.9–2.9) |
| Kawolo Hospital | Before | 18 | | | | |
| | After | 53 | 0.005 | 2.9 | <0.001 | (2.3–3.6) |
| Kiwoko Hospital | Before | 4 | | | | |
| | After | 6 | 0.223 | 1.6 | 0.105 | (0.9–5.4) |

* **Independent samples** t-Test

170). Iganga hospital had an increase of 3(from 6 to 9). Kawolo hospital had an increase of 16 (from 67 to 83). Kiwoko hospital registered a decline at -16(from 83 to 67) as shown in Figs 1 and 2 below.

## Discussion

This research project sought to resolve the gap in adolescent TB care seeking behavior using a human centered approach. This was an approach where a contracted designing company, in this case, "Design without borders (DwB)", involved and worked together with the adolescents to develop a TB awareness and screening package. The package consisted of a local song ("*Bulamu bwo*"), TB poster messages and Screening cards. The local song targeting adolescents with a TB message was broadcasted in the community radios. Poster messages were displayed in adolescent meeting points in the communities within the catchment areas of the project health facilities. The TB screening cards were given to TB positive or presumptive adults in the clinics who had adolescents at home. The cards were given to the adults after they had been

**Table 5. Incident rate ratio (IRR) of adolescents tested for TB.**

| Facility | Period | Mean* | | Likelihood | | |
|---|---|---|---|---|---|---|
| | | Mean(# Tested) | p-value | IRR | p-value | 95% CI |
| **Overall** | **Before** | 8 | | | | |
| | **After** | 28 | 0.007 | **3.3** | <0.001 | (2.8–3.8) |
| **Gombe** | **Before** | 6 | | | | |
| | **After** | 10 | 0.102 | 1.7 | 0.012 | (1.1–2.7) |
| **Iganga** | **Before** | 7 | | | | |
| | **After** | 42 | 0.072 | 6.5 | <0.001 | (4.6–9.1) |
| **Kawolo** | **Before** | 18 | | | | |
| | **After** | 53 | 0.005 | 2.9 | <0.001 | (2.3–3.6) |
| **Kiwoko** | **Before** | 4 | | | | |
| | **After** | 6 | 0.223 | 1.6 | 0.105 | (0.9–2.7) |

* Independent samples t-Test

oriented by health workers and project volunteers on proper use. The adults then took these cards home and screened their adolescents. Presumptive adolescents found at home were referred by their caretakers to project health facilities for further TB screening and testing. These three interventions were piloted for six months (from October 2021 to March 2022). Adolescents who came to the health facilities were further screened for TB. All those found with one of the TB symptoms, eighteen years and above were consented and enrolled. Those less than eighteen years were enrolled after receiving ascents from their caregivers. Enrolment was done in the four project health facilities in central Uganda. After enrolment, each participant was taken through the entire TB cascade. Before and after data of adolescent TB enrolment was obtained in all the four project health facilities and analysed. Generally findings indicated a statistically significant increase in the average number of adolescents screened, presumed and tested for TB. This was evidenced by the increase in average number of adolescents screened from 159 before to 309 after the intervention, an increase that was statistically significant (p <0.001), The average number of those presumed increased from 13 before to 29 after the intervention, and this increase was also statistically significant (p <0.001). The average numbers tested increased from 8 before to 28 after the intervention, and likewise this was statistically significant (p <0.001). This research project was carried out at a time when there was COVID-19 outbreak, however, despite our findings, a study that looked at "integration of COVID-19 and TB screening in Kampala, Uganda: "healthcare provider perspectives"; showed a 40% drop in TB screening following the first wave of COVID-19 outbreak in 2020 [8]. Besides the increase in average number of cases screened, presumed and tested, our project registered a minimal number of TB positives that were linked to care. The minimal numbers could be attributed to improved TB service delivery, treatment coverage (84%) and treatment success rate (87.3%) in Uganda (2020/1) [9]. However, despite the few numbers of TB positives registered, our intervention mobilized many (394) of adolescents to seek TB care in all the project health facilities. Waako and his colleagues in a study they conducted in eastern Uganda on the burden of TB disease among adolescents emphasized contact tracing as a means to increase case detection among adolescents [10]. Our research Project explored adolescent contact tracing, where TB positive and presumptive adults screened their own adolescents at home, and referred the presumptive to the health facility for further TB screening and testing. Another study conducted in Uganda on delay in diagnosis and treatment of TB suggested the need for TB advocacy in the communities [11]. In regard to TB community advocacy, our project explored the use of a local song broadcasted in community radios stations and poster messages deployed in the communities to create TB awareness.

Generally our research project enrolled more Female adolescents; 255(65%), than males. This finding was quite similar to what was indicated in the Uganda national TB prevalence survey 2015, in which; men with TB symptoms were seen to seek health care less often than women; suggesting existence of disparities, both geographical and gender for the TB situation in the country. While the survey attributed the TB health care behaviour to disparities, another adolescent study attributed the high number of females who come for TB care to their high susceptibility to TB as compared to their male counterparts [12]. Forty nine (68%) of the adolescents reported having TB contact with family members, close contact mixing is a common phenomenon demonstrated at household settings. A study that looked at age-and sex-specific social contact in the Zambian and South African communities revealed that more than 50% of TB infections in children resulted from contacts with adult men [13].

Majority of the enrolled adolescent had no history of smoking or alcohol consumption. Cigarette smoking and alcohol consumption in this research project were considered as minimal risk factors for adolescent TB. However, regardless of the above finding, cigarette smoking

and alcohol drinking continue to be among the first five top risk factors associated with TB globally [14].

Among the key TB awareness interventional approaches that were implemented to close the gap in adolescent TB care seeking, the local song ("*Bulamu bwo*") broadcasted on radio stations sensitized more adolescents to seek care as compared to TB adolescent poster messages and TB screening done at house hold level by parents or caregivers. Just like this project utilized an awareness strategy, a study in Bangladesh similarly emphasised on increased awareness and service delivery by health care workers as promoters and motivators of increased health care seeking behaviour [15].

Besides the three key intervention approaches, many received information about TB from the research teams in the health facilities, community village health teams (VHTs) and social media. This tends to suggest that the awareness that was created by the intervention, motivated the adolescents to come to the health facilities.

Screening done by parents or guardians at household level demonstrated a potential community approach to reach and mobilize adolescents to seek care. This kind of intervention is a way to enhance contact tracing. A study done by Hanrahan CF et al. documented contact tracing as an approach to easily reach and mobilize TB presumptive patients for testing [16].

Most of the adolescents that were enrolled into the project had prior knowledge on TB and had ever heard about adolescent TB, this was an encouraging finding because lack of knowledge has been alluded to by Owolabi O.A and his colleagues as a leading barrier to TB health care seeking [17].

## Limitations

Being a pilot project implemented at the health facilities, we were not able to reach out to other adolescent communities such as schools and distant villages. These results are unlikely to be generalizable. Secondly, our study depended on self- reported responses, which could be affected by recall bias. Out of the several interventional methods that were provided by the expert designers, we were only able to implement three due to COVID-19 outbreak and restrictions. This could have had a negative impact on the number of adolescents reached. Additionally, it was not possible to come up with appropriate comparisons of the three interventions since they were not implemented at the same time, though conclusions were drawn after harmonizing the intervention periods.

## Conclusion

The intervention generally improved adolescent TB care seeking behavior along all the important steps of the TB care cascade (screening, TB testing, TB positives and linkage to care). The use of human centered approach that directly involved the adolescents in initial development of the interventions was a real strength for the project success. Hence, there is great need to devise mechanisms to mobilize adolescents to seek TB care. Since this was a pilot project, we recommend a robust and fully powered randomized controlled trial to mobilize more adolescents and more so evaluate the effectiveness of the package.

## Supporting information

**S1 Abstract.**
(XLSX)

**S1 File.**
(PDF)

**S1 Data.**
(XLSX)

**S1 Raw data.**
(XLSX)

## Acknowledgments

The entire implementation process and success of teenager's TB project is attributed to the great contribution of different stakeholders. We specifically thank the national TB and leprosy program administration (MOH), which provided administrative clearance and technical support at national level. Design without Borders (DwB) Africa limited team, a creative design group which developed the adolescent TB screening cards & poster messages. These were used for household screening and TB awareness creation respectively. We as well extend our sincere appreciation to the "*Ghetto Yute*" youth music group, which produced the local song ("*Bulamu bwo*") with adolescent TB awareness messages. The song was broadcasted in selected local radio stations, within the surrounding communities of the project health facilities. Additionally, we would want to thank the health workers who worked closely with research assistants, village health teams (VHTs) and project volunteers to collect data in the health facilities. Last but not least, the Government of Uganda and Makerere University.

## Author Contributions

**Conceptualization:** Samson Omongot, Winters Muttamba, Irene Najjingo, Sabrina Kitaka, Bruce Kirenga.

**Data curation:** Samson Omongot, Irene Najjingo.

**Formal analysis:** Samson Omongot, Winters Muttamba.

**Funding acquisition:** Bruce Kirenga.

**Investigation:** Samson Omongot, Winters Muttamba, Irene Najjingo, Joseph Baruch Baluku.

**Methodology:** Samson Omongot, Winters Muttamba.

**Project administration:** Samson Omongot.

**Resources:** Bruce Kirenga.

**Software:** Winters Muttamba.

**Supervision:** Winters Muttamba, Sabrina Kitaka, Stavia Turyahabwe, Bruce Kirenga.

**Validation:** Winters Muttamba.

**Visualization:** Samson Omongot, Joseph Baruch Baluku.

**Writing – original draft:** Winters Muttamba, Irene Najjingo, Sabrina Kitaka, Stavia Turyahabwe, Bruce Kirenga.

**Writing – review & editing:** Samson Omongot, Winters Muttamba, Irene Najjingo, Joseph Baruch Baluku, Sabrina Kitaka, Stavia Turyahabwe, Bruce Kirenga.

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
