## [Decision Letter · Decision Letter 0]

23 Aug 2023

PONE-D-23-15403Strategies to resolve the gap in Adolescent Tuberculosis care at four health facilities in Uganda: The TEEN TB pilot projectPLOS ONE

Dear Dr. Omongot,

Thank you for submitting your manuscript to PLOS ONE. After careful consideration, we feel that it has merit but does not fully meet PLOS ONE’s publication criteria as it currently stands. Therefore, we invite you to submit a revised version of the manuscript that addresses the points raised during the review process.<ol><li> 

Please try to strictly attend the PLOS ONE submission criteria-style of writing up the title, authors list, text referencing, table situation,<li> 

Check the way headings and subheadings are narrated. Also, the structure and outline of the whole manuscript must be aligned with THE PLOS ONE format.

https://journals.plos.org/plosone/s/submission-guidelines#loc-cell-lines

3.0 Please seriously attend to the structure, grammar, and punctuation in the manuscript-- highly recommended to review the languages of the manuscript.

4.0 Please try to debulk the methods section and try to be brief. 

5.0 Also. try to justify special reason to undertake your study--comparing and referring to previous evidence on Adolescent TB in Uganda

6.0 Strongly recommend describing the peculiar finding of your study.

A rebuttal letter that responds to each point raised by the academic editor and reviewer(s). You should upload this letter as a separate file labeled 'Response to Reviewers'A marked-up copy of your manuscript that highlights changes made to the original version. You should upload this as a separate file labeled 'Revised Manuscript with Track Changes'.An unmarked version of your revised paper without tracked changes. You should upload this as a separate file labeled 'Manuscript'.

We look forward to receiving your revised manuscript.

Kind regards,

Zewdu Gashu Dememew, M.D, PhD

Academic Editor

PLOS ONE

Journal Requirements:

"NO"

"The authors  declare  no competing interests."

Reviewers' comments:

Reviewer's Responses to Questions

**Comments to the Author**

1. Is the manuscript technically sound, and do the data support the conclusions?

Reviewer #1: Partly

Reviewer #2: Yes

Reviewer #3: Yes

Reviewer #4: Partly

2. Has the statistical analysis been performed appropriately and rigorously? 

Reviewer #1: Yes

Reviewer #2: Yes

Reviewer #3: Yes

Reviewer #4: No

3. Have the authors made all data underlying the findings in their manuscript fully available?

Reviewer #1: Yes

Reviewer #2: Yes

Reviewer #3: Yes

Reviewer #4: No

4. Is the manuscript presented in an intelligible fashion and written in standard English?

Reviewer #1: No

Reviewer #2: Yes

Reviewer #3: Yes

Reviewer #4: Yes

5. Review Comments to the Author

Reviewer #1: Although the study has merit it fails to add to the existing knowledge base in this area. There have been numerous studies that have been published in this area of research (covering broader aspects) and from similar settings. For example:

Muttamba W, Bbuye M, Baruch Baluku J, Kyaligonza S, Nalunjogi J, Kimuli I, Kirenga B. Perceptions of Adolescents and Health Workers Towards Adolescents’ TB Diagnosis in Central Uganda: A Cross-Sectional Qualitative Study. Risk Management and Healthcare Policy. 2021 Nov 30:4823-32.

Ssengooba W, Kateete DP, Wajja A, Bugumirwa E, Mboowa G, Namaganda C, Nakayita G, Nassolo M, Mumbowa F, Asiimwe BB, Waako J. An early morning sputum sample is necessary for the diagnosis of pulmonary tuberculosis, even with more sensitive techniques: a prospective cohort study among adolescent TB-suspects in Uganda. Tuberculosis research and treatment. 2012 Dec 4;2012.

Maleche-Obimbo E, Odhiambo MA, Njeri L, Mburu M, Jaoko W, Were F, Graham SM. Magnitude and factors associated with post-tuberculosis lung disease in low-and middle-income countries: A systematic review and meta-analysis. PLOS Global Public Health. 2022 Dec 20;2(12):e0000805.

Reviewer #2: Review Reports

Title: Strategies to resolve the gap in Adolescent Tuberculosis care at four health facilities in Uganda: The TEEN TB pilot project

Manuscript Number: PONE-D-23-15403

Review Comments

The title can be resentenced

The background is inadequate and needs strength for its contents.

The abstract is too long.

Why you have used both the P value and the confidence interval at the same time? The result section fails to compare the before after results of the intervention.

The manuscript fails to meet the standard guideline of PONE E.g. The study setting come after the study design

What is the specific study design used?

Did the diagnostic capacity or positive predictive rate of t both Gene expert and smear microscopy equal? Why you have used either of the test equally otherwise?

What type of specific consents were used for those less than 15, 15-18 and greater than 18 years?

How do you differentiate that the adolescents had arrived to the community due to your particular intervention?

What are the specific interventions? What is the dose, schedule, the information conveyer quality? Means of delivery? Cultural appropriateness and language of the intervention needs detail explanation?

What was done to ensure data quality?

What does descriptive univariate analysis mean?

The results, discussion, conclusion and recommendation needs major revision in line with the objective.

Regards,

Reviewer #3: Dear Author.

Thank you for the above study.

I must say that it is very relevant and well conducted by your team.

Due to the period in which the study was conducted and the prevailing epidemiology, I think your study was too quiet about Covid-19 impact. It affected your results which I was expecting you to rigorously discuss in the discussion section.

I think the impact of covid 19 is beyond the restriction of movement as you mentioned in the conclusion section. The periscope of infectious diseases cannot be turned away from Covid at the time when your study was conducted... there is a lot of studies on this: TB and Covid-19.

My OTHER COMMENTS ARE AS BELOW: please attend to them as in the attached document:

Thank you.

Strategies to resolve the gap in Adolescent Tuberculosis care at four health facilities in

Uganda: The TEEN TB pilot project.

Consider a short title that captures the attention of the reader e.g. " Adolescent Tuberculosis care in Uganda: Participatory action research"

This is a well written abstract. The chronology and clarity of the content are commendable. Thank you.

Overall, this is a good introduction with a precise expression of the social values of the study and a focused scientific values . please address the comment below

Thank you for this comparative epidemiology. I think it would have been better comparing 2021 with a year before the Covid -19 pandemic.

The reason for this is obvious from infectious disease perspective.

Please look into this. Data from 2018 could be better. thanks

Are you referring to TB notification surveillance?

If not, please substantiate your comment.

If yes, this comment is incorrect. TB notification exists as part of the standard of care of TB as an infectious communicable disease for every country under WHO.

I think you need to talk briefly about these study sites. why are they important to this study? what is the basis for their selection as a study site?

Good description of the study sites. Ignore the question about this above

Rather use: Unintentional weight loss or noticeable unintentional weight loss or just " weight loss"

Good description of the data collection procedure. Thanks

Good description of the steps in the data analysis

It is good that the clinical characteristic question addressed the knowledge about HIV due to its relevance in TB disease. I would have expected the same knowledge to be explored wrt Covid-19, due to its relevance at the period of your study and its impact on TB during the pandemic.

I think you may need to defend the clinical significance of this in the "discussion section". The fact that it is statistically insignificant (P= 0.170) doesn’t necessarily means that it is the same clinically.

I am of the opinion that you should separate the " Limitation of the study" from the conclusion.

Most of what you mentioned here are limitations.

Reviewer #4: I would like to congratulate the authors for this important work. This project may be an important input to the National TB and Leprosy Programs to strength the effort to control, prevent, and eliminate TB. Having said this, the study needs major revision that needs to be revised carefully.

Specific comments

Abstract

• It is not clear how the total study participants were 394 in fact the average number of adolescents screened increased by 94% from 159 to 309 that gives a total of 394. Please clarify in detail. If it was a daily/monthly average of the four sites? Specially describe the detail in the methods section.

• What type of randomized control trial? Is it cluster/population/community based or individual based?

Introduction

• No comment

Methods

• Describe who collected the socio-demographic and clinical data

• In the project settings section;

o Describe how the four health facilities are selected from the whole health facilities in the country? Please justify it. It is not realistic to randomly select four hospitals across the health facilities in the country. Two were from urban and two from rural, it seems purposively selected.

o Are all four health facilities are hospitals? You described that the study included health centers and district level hospitals? But all four were hospitals.

• In the data collection section;

o You described an idea already described in the previous section “Adolescents (10-19 years) who came to the project health facilities were identified from the service delivery points (OPD, ART clinic, MRH clinic, TB clinic and wards) and screened for TB using the national screening algorithm. Those with at least one of the TB symptoms and not on anti-TB treatment were considered eligible and were consented (≥ 18years) or assented (≤ 18 years) by their parents or health workers and enrolled into the project.”

o Describe about how data quality is maintained? Please describe one paragraph about data quality?

o How did you control if one individual came before and after intervention?

• In the data management/analysis section

o Change bivariate analysis to bi-variable. Univariate analysis includes bi-variable and multi-variable analysis. Bivariate analysis is used when analyzing more than one dependent variables at the same time.

o Describe which T-test was used (independent vs paired)

• Include operational definitions

Result

• Under the clinical characteristics section;

o I prefer to report in the opposite, such that how many smoked.

o 60(15%) of these reported HIV positive status which is a big proportion.

o To calculate the proportion of the participants who reported to have information from the research team, the denominator should be those who heard (326). 266/326=81.6%

o Do all participants visited the health facilities for TB care service? If yes, how many?

• Under the screening for TB section; line 5-10

o It is not clear about the number screened for TB. The sum of the number screened in the four hospitals is bigger than your sample size. Is it about the general population or adolescents who screened for TB? You described that the number of adolescents increased from 159 to 309=468 total adolescents, but the figures talk about 394 adolescents. Please clarify this. This works for all sub sections in the result section. Please early clarify in the methods section or early result section.

• Under the TB presumptive patients section;

o At line two and three, you described that the overall average increase after the intervention was from 13 to 29. However, based on the data from each hospital the sum of all the four hospitals is big number. Generally, the numbers in all the results section including TB testing , TB positive and linkage sections are ambiguous, please clarify in detail

• Under the linkage to TB care and treatment section;

o You described there was 50% (2 to 3.1) increase. How the number of adolescents can be described in decimals (3.1). Is it a mean?

Figures

• What mean by mean? E.g. Mean of screened?

Tables

• Table 1:

o The sum of the frequencies in each hospital is not equal to the overall frequencies. 159 vs 224+213+185+15? Are these figures talks about adolescents or general population? This applies for all tables and figures. Make it clear.

Discussion

• General;

In many sentences of the discussion section, you described that many studies support the study finding of your study. However, you cited only one reference that needs to be revised. E.G., paragraph two about males vs females, paragraph three about close contact, paragraph four about cigarrete smoking. Since line numbering is not available, difficult to put minor comments specific to the line number.

• Paragraph two you described that more female adolescents participated compared to males and tried to link to a previous study (ref 11). However, your justification may not be correct because among the global TB cases, males are predominant. Better to suggest the possible reason.

• In paragraph three you described majority of the TB positives were students, and close contacts, but I didn't see any result in tables, figures or narratives that described the link of TB positivity with any risk factors/variables in the result section. You have to focus on the main objectives or describe these findings in the result section. Discussing something you did not described in the result section or in the supplementary files is inappropriate. The objectives, the results and discussion should talk about similar things. This works also for cigarrete smoking. Have you done regression analysis or chi-squared test? If yes, what was the p-value?

• You described that some studies have shown close association between TB and cigarrete smoking alcohol consumption. However, you cited only one reference. In addition, why only some studies? Both factors are the known risk factors for TB and included in the five top factors contributing for TB in the Global TB report. Please revise it.

• You discussed about x-ray results. Please describe all parameters in the result section. Then you can discuss it.

• The paragraph “Besides the three key intervention approaches, many received information about TB from the research teams in the health facilities, community village health teams (VHTs) and social media. Apart from lack of information, other similar studies have also documented stigma, delays in diagnosis, long waiting hours, absence of nutritional support and psychosocial support as related factors associated with poor TB health care seeking 8 , 17” is not covered in your study. I suggest removing this paragraph.

• Again, you described many studies/ other studies while citing only one study. Please revise it.

• Please discuss more on the objectives of the study. Focus on the objectives of the study. Discuss more on the intervention model and the results before and after the intervention.

• Is there a statistical difference among the hospitals, among rural and urban hospitals? Please did analysis on this issue, describe in the result section and then discuss if appropriate.

• Please bring the limitations described in the conclusion section at the end of the discussion section in one paragraph. Because the readers should understand the limitations before go to the conclusion.

• Please use similar referencing style. Vancouver vs Harvard?

• Based on the things described in the result section about risk factors for TB, there is data gap not described either with in the manuscript or the supporting information.

Minor comments

• Include line numbering

• In the Abstract section, change Introduction to Background

• In all sections, describe the numbers up to 10 in the full name/words. E.g., change 4 to four.

• In the result of the abstract section, change GeneX-pert to GeneXpert.

• In the data management/analysis section fourth line change the data was cleaned to the data were cleaned.

6. PLOS authors have the option to publish the peer review history of their article (what does this mean?). If published, this will include your full peer review and any attached files.

Reviewer #1: No

Reviewer #2: No

Reviewer #3: **Yes: **Adeloye Amoo Adeniji (MBBS; MMed; FCFP; FACRRM)

Reviewer #4: No

---

## [Author Response · Author response to Decision Letter 0]

11 Nov 2023

PONE-D-23-15403: Responses to the review comments 

Comment: Consider a short title that captures the attention of the reader.

Response: The title has been shortened to read “Strategies to resolve the gap in Adolescent Tuberculosis care in Uganda: The TEEN TB pilot project” as indicated in line No. 1&2 on the manuscript 

Comment: The abstract is too long.

Response: It has been shortened in the revised manuscript- refer to line Nos. 16 to 49 in the Manuscript 

Comment: The background is inadequate and needs strength for its contents.

Response: Additional literature on Adolescents Tuberculosis has been added- refer to the main background on the manuscript- No 51 -81

Comment: Why you have used both the P value and the confidence interval at the same time?

Response: The P Values and confident intervals have been re- arranged in the result section of abstract and main manuscript On line no 35 and 176 respectively in the manuscript 

Comment: The manuscript fails to meet the standard guideline of PONE e.g. the study setting came after the study design

Response: The order was changed in the revised Manuscript, i.e., we have the study setting before study design; refer to.line number 83-92 & 93-132 in the Manuscript

Comment: What is the specific study design used?

Response: It was a human centered design- Meaning the adolescents were involved in designing process of the intervention package (i.e. the TB awareness posters, screening cards and the local song) at the initial stage. This is clearly explained in the project design section; line no 94-96 of the manuscript

Comment: Did the diagnostic capacity or positive predictive rate of both Gene Xpert and smear microscopy equal? Why you have used either of the test equally otherwise?

Response: We deleted the word” smear microscopy” because more than three quarters of the samples were tested using Gene Xpert, the few that were tested microscopically were removed during data analysis.

Comment: What type of specific consents was used for those less than 15, 15-18 and greater than 18 years?

Response: We used Assent forms for adolescents’ less than 18 years and Consent forms for those who were 18 years and above. It’s clearly explained in the methodology section; line no 124 in the manuscript.

 Comments: How do you differentiate that the adolescents had arrived to the health facility due to your particular intervention?

Response: For the screening cards, the ones taken and returned from the community were tracked using a health facility tracker. For the TB awareness posters and the local song, a TB assessment form was used to find the source of information before they were enrolled to the project.

Comment: What are the specific interventions? What is the dose, schedule, and the information conveyer quality? Means of delivery? Cultural appropriateness and language of the intervention needs detail explanation?

Response: 1- Tuberculosis screening cards with TB case finding questions(English & local language) were used by the TB positive adults to screen their own adolescents at home, any of the adolescents that were found with one or more of the TB symptoms were referred to health facilities for further assessment and care.- No 115 – 119 in the Manuscript

2- Tuberculosis awareness posters (English & local language) were designed and deployed in communities within the catchment areas of the project health facilities. These were mostly placed in locations frequently accessed by adolescent like markets, places of worship and drug shops- No 112 and 113 in the manuscript.

These were deployed one month before the study begun and throughout the study period.

3- The Local song was played from selected radio stations with wider coverage to the targeted communities. The song was played for one month before the study begun and throughout the study period- No 114 and 115 in the manuscript.

Comment: What was done to ensure data quality?

Response: A qualified Research team was trained on the study protocol, data collection tools were designed, approved by the ethics committee and pre-tested before data collection. The Kobo collect software was used to collect the data, the data was cleaned and queries resolved before transportation to STATA version 14 for analysis. Please refer to the section on data quality control (line no. 143- 148) incorporated on the revised manuscript

Comment: What does descriptive univariate analysis mean?

Response: This was a mistake in writing; we meant descriptive statistics was used to summarize and describe individual study variables- refer to line no 163 in the revised manuscript

Comment: The results, discussion, conclusion and recommendation need major revision in line with the objective.

Response: Revision was done for discussion, conclusion and recommendations sections Limitations were separated from the conclusion; refer to the attached revised manuscript line no. 336, and 429 respectively in the manuscript

Comment: Please clarify the sources of funding (financial or material support) for your study.

Response: The study was funded by Makerere University research and Innovation fund (MAKRIF).

Comment: State what role the funders took in the study.

Response: The funders had no role in study design, data collection and analysis, decision to publish, or preparation of the manuscript.

Comment: If any authors received a salary from any of your funders, please state. 

Response: None of the authors received a salary from the funders.

Comment: In your Data availability statement, you have not specified where the minimal data set underlying the results described in your manuscript can be found?

Response: All relevant data and the supporting information files have been provided

Comment: If you did not receive any funding for this study, please state

Response: The authors received funding for carrying out research, but no financial support to authorship and publication of this manuscript.

NB. All minor comments (structure, grammar punctuations), and all suggestions by the editor and reviewers were catered for and reflected in the revised manuscript.

---

## [Editor Report · Decision Letter 1]

28 Nov 2023

PONE-D-23-15403R1Strategies to resolve the gap in Adolescent Tuberculosis care at four health facilities in Uganda: The TEEN TB pilot projectPLOS ONE

Dear Dr. Omongot,

Thank you for submitting your manuscript to PLOS ONE with some comments addressed. After careful consideration, we feel that it has merit but does not fully meet PLOS ONE’s publication criteria as it currently stands. Therefore, we invite you to submit a revised version of the manuscript that addresses the points raised during the review process.

We look forward to receiving your revised manuscript.

Kind regards,

Zewdu Gashu Dememew, M.D

Academic Editor

PLOS ONE

Journal Requirements:

**Additional Editor Comments:**

Dear Authors,

Thank you again for the revision,

However, there are remaining unaddressed issue to your attention. Hope you will attend to these commnest

Regards

---

## [Author Response · Author response to Decision Letter 1]

16 Jan 2024

PONE-D-23-15403R1: Response to reviewer’s comments.

1-Comment: Title: replace ‘Adolescent Tuberculosis’ with ‘adolescent tuberculosis’

Response: The replacement was done as indicated on line number 1 of the manuscript.

2-Comment: Abstract; you may omit the headings-- introduction, methodology, results, and conclusions.

Response: These heads were removed; refer to the abstract on line numbers 17-47 of the manuscript.

3-Comment: Please write the full name of TEEN

Response: The full name for ‘TEEN’’ is “teenager’s” as indicated on line number 2 and 422(acknowledgement) of the manuscript.

4-Comment: Methods; Note: the study/research design is before and after or quasi experiments while the framework or a study approach is human centered design (HCD).

Response: The sentences were re-written as indicated on line numbers 23 &27(abstract) and 93 & 96-97 (project design) - refer to the manuscript.

5-Comment: “There was a minimal increase in the average monthly number of adolescents with a positive result from 1.6 to 2.4 and linkage to TB care services from 2 to 3.1. These were not statistically significant at p=0.170 and p=0.154 respectively.” Could be replaced with “There was a minimal increase in the average monthly number of adolescents with a positive result from 1.6 to 2.4(p=0.170) and linkage to TB care services from 2 to 3.1 (p=0.154)”

Response: The correction was made as indicated in line numbers 43 & 44 of the manuscript

6-Comment: Introduction- Can you make the data source of TB epidemiology of global, Africa and Uganda same years—preferably 2022 global TB report.

Response: The Global, African and Ugandan TB epidemiology data for adolescents aged five to fourteen years was considered for the same year (2021) as reflected in 2022 global report. This is indicated on line numbers 53 to 61 of the manuscript.

7-Comment: # less than 10 could be written in words e.g. 4- four, 8=eight etc. No need to write like three (3), four (4), and six (6)…

Response: The numbers were written in words thus; fourteen, in line number 19, four as shown on line numbers 99 , three as indicated in line number 106 & 121, six on line 120 of the manuscript.

8-Comment: Please avoid capital letters in the middle of the sentence e.g ‘’The training focused on Four (4) key interventions…’’ can be written as ‘‘’The training focused on four key interventions…’’

Response: This was corrected as the training focused on four key ingredients, as indicated in line number 107 of the manuscript.

9-Comment: Check the consistence of clinical criteria.“… those who presented with any of the four symptoms (cough, fever, drenching night sweats and weight loss) were consented (≥ 18years), or assented (< 18 years) and enrolled into the project.” 

AND 

“… Predefined according to the World Health Organization’s (WHO) criteria (cough for ≥2 weeks; persistent fever for ≥2 weeks; noticeable weight loss; and excessive night sweats).”

Response: The sentence was re-written as; “Those who presented with predefined tuberculosis symptoms according to the World Health Organization’s (WHO) criteria (cough for ≥2 weeks; persistent fever for ≥2 weeks; noticeable unintentional weight loss; and excessive night sweats), were consented if they were eighteen years or more, or assented if less than eighteen years. All those who consented or assented were enrolled into the project.”

Refer to line numbers 122 to 127 of the manuscript.

10-Comment: Write the full name of OPD. As first appeared all abbreviations should be written in full. E.g. Anti-retroviral therapy (ART), outpatient department (OPD)….

Response: These have been corrected and written in full names; please refer to line number 137 & 138 in the manuscript.

11-Comment: Replace ‘’ Data management/analysis’ with ‘’ Data management and analysis’’.

Response: “Data management /analysis procedure” was written as” Data management and analysis”, refer to line number 164 in the manuscript.

12-Comment: ‘Ethics approvals and informed consent’ is usually written as ‘Ethical consideration or Ethical review”. Please check the the PLOS ONE journal criteria.

Response: ‘Ethics approvals and informed consent’, was re- written as “Ethical considerations”, “Ethical approvals”, was re- written as “The research project ethical approvals”; Please refer to line number 172 & 173 respectively in the manuscript.

13-Comment: Result; Please re-check punctuation. Example

‘’Majority, 198 (50%) were aged between 10-15 years, they were mainly females 255(65%), and still in school 298(76%). Majority, 336(85%) were unmarried. Kawolo hospital enrolled the highest number, 137(35%)’’ could be written as ‘’ Majority, 198 (50%), were aged between 10-15 years. They were mainly females 255 (65%), and still in school 298(76%). The majority, 336(85%), were unmarried. Kawolo hospital enrolled the highest number, 137(35%’’

Response: Corrections were made on punctuation; as indicated on line numbers 184 - 187 in the manuscript.

14-Comment: Table 1: Check for the age group, includes 20 years

Response: The age group (16- 20years) on table 1 was corrected to (16- 19 years). Refer to table 1 on line number 191 in the manuscript.

15=Comment: Up to 330 (84%) can be written as ‘‘Three hundred thirty (84%)…’’

Response: The sentence was reframed as sixty four (16%) of adolescents had never heard about TB, as indicated on number 196 of the manuscript.

16-Comment: Up to 49 (68%) can be written as ‘’ Forty nine (68%)…’’

Response: The sentence was re-written as shown on line number 196 of the manuscript. All other sentences with similar issues were re-written as presented on line numbers 196- 203.

17-Comment: You may delete ‘’ The impact of the package on adolescent TB care is presented along the TB care cascade In Figure 1, 2, Tables 3, 4 and 5. The entire analysis revealed the following results:’’

Response: Response: The sentences were deleted as indicated on line numbers 254, 255 & 256 of the tracked manuscript. In the manuscript, line number 209 has the heading “Impact of package on TB care cascade”, indicated after the rest of the words were deleted.

118-Comment: Reference: please correct per the PLOS ONE style throughout- not in superscript1 but in bracket [1]. The list of references should be per the PLOS style, Vancouver.

Response: The references that were written as superscript have been written in bracket format in line with Vancouver style, refer to line numbers 52,53,57,59,61,63.65,68,70,74,and 76 (Background), and 348, 351, 356, 360, 370, 374, 379, 386, 395, and 399(Discussion) in the manuscript.

NB. 1- The former references (no. 2 and 3) were deleted; reference numbers from 4 to 19 in the tracked manuscript were replaced with 2 to 17 respectively as indicated in the manuscript.

NB. 2- Page numbering was revised in the tracked manuscript and manuscript. 

NB. 3- Minor corrections, e.g. on punctuations, spellings, wording, etc., have been attended to in the tracked manuscript and the manuscript.

---

## [Editor Report · Decision Letter 2]

1 Feb 2024

Strategies to resolve the gap in adolescent tuberculosis care at four health facilities in Uganda: The Teenager's TB pilot project.

PONE-D-23-15403R2

Dear Authors,

We’re pleased to inform you that your manuscript has been judged scientifically suitable for publication and will be formally accepted for publication once it meets all outstanding technical requirements.

Kind regards,

Zewdu Gashu Dememew, M.D, PhD

Academic Editor

PLOS ONE

---

## [Editor Report · Acceptance letter]

3 Apr 2024

PONE-D-23-15403R2 

PLOS ONE

Dear Dr. Omongot, 

I'm pleased to inform you that your manuscript has been deemed suitable for publication in PLOS ONE. Congratulations! Your manuscript is now being handed over to our production team.

Kind regards, 

on behalf of

Dr. Zewdu Gashu Dememew 

Academic Editor

PLOS ONE